# A Close Look At World Model Recovery in Supervised Fine-Tuned LLM Planners

## Abstract

Supervised fine-tuning (SFT) improves end-to-end classical planning in large language models (LLMs), but do these models also learn to represent and reason about the planning problems they are solving? Due to the relative complexity of classical planning problems and the challenge that end-to-end plan generation poses for LLMs, it has been difficult to explore this question. In our work, we devise and perform a series of interpretability experiments that holistically interrogate world model recovery by examining both internal representations and generative capabilities of fine-tuned LLMs. We find that: a) Supervised fine-tuning on valid action sequences enables LLMs to linearly encode action validity and some state predicates. b) Models that struggle to use output probabilities for classifying action validity may still learn internal representations that separate valid from invalid actions. c) Broader state space coverage during fine-tuning, such as from random walk data, yields more accurate recovery of the underlying world model. In summary, this work contributes a recipe for applying interpretability techniques to planning LLMs and generates insights that shed light on open questions about how knowledge is represented in LLMs.

## 1 Introduction

Reliable automation of complex workflows with large language models (LLMs) depends on their ability to successfully generate plans. LLMs that plan can help humans write software, recommend contingencies in the face of emergencies, and contribute to open-ended scientific discovery. However, current pretrained LLMs have significant difficulty with generating plans in zero-shot and few-shot settings (Valmeekam et al., 2024). Performance decreases as problem complexity increases beyond what is seen during training and LLMs hallucinate intermediate steps of plans even when arriving at the goal (Momennejad et al., 2023). This has driven interest in fine-tuning to improve the planning abilities of LLMs (Pallagani et al., 2022; Hirsch et al., 2024). There is reason for optimism; supervised fine-tuning (SFT) and reinforcement learning (RL) fine-tuning, the two major LLM fine-tuning paradigms, have been used to greatly enhance the reasoning capabilities of LLMs in other domains, such as math problem solving and coding abilities. LLMs explicitly fine-tuned to solve difficult problems that require deliberation show noticeable improvements over their non-reasoning counterparts when evaluated on planning benchmarks (Valmeekam et al., 2025).

An important question is whether LLMs that are effective planners after fine-tuning are also representing and reasoning about planning problems. One way in which LLMs might reason about a planning problem is by using a world model to decide which actions to take while forming a plan. There is evidence that suggests that LLMs do not natively reason this way about planning problems (Hirsch et al., 2024). However, a popular hypothesis is that, under favorable training conditions, LLMs can *learn* to use world models to make predictions. Recent work (Toshniwal et al., 2022; Li et al., 2022) studies world model recovery in board-game-playing transformers, equating recovery in Othello and Chess with the identification of internal representations that encode the game's state.

Compelling evidence that LLMs may be recovering world models *to some extent* is now accruing. For example, they appear to learn internal representations of meaning that are updated during dialogue (Li et al., 2021), as well as linear representations of numbers (Kadlčík et al., 2025) and even their own hallucina-

tions (CH-Wang et al., 2023). Xie et al. (2024) shows that for a synthetic logic puzzle domain, LLMs learn to reason from fine-tuning while also learning interpretable representation essential to solving the puzzles. There are previous studies which suggest a different perspective: transformers are liable to learn shortcut solutions to automata (Liu et al., 2022; Vafa et al., 2024) and logical reasoning problems (Zhang et al., 2022), motivating our close look in the classical planning setting.

In this work, we raise the following question in the context of classical planning: *Does SFT with examples of plans induce world model recovery in LLMs?* We adapt two popular definitions of world model recovery (Toshniwal et al., 2022; Vafa et al., 2024):

> 1. **Internal representations**: We say that a fine-tuned LLM has recovered a planning world model if it has learned representations that encode the truth values of state predicates and validity of actions in states.
>
> 2. **Generation**: We say that a fine-tuned LLM has recovered a planning world model if it only assigns high probability to sequences of valid actions and low probability to sequences containing invalid actions.

Our paper develops a framework for studying world model recovery under these definitions in classical planning, and contributes new insights to this ongoing debate by fine-tuning a collection of `gemma2-9b-instruct` (Team et al., 2024) models with SFT on end-to-end plan generation. To quantify world model recovery from the lens of **internal representation** quality (1), we train linear probes on the hidden states of fine-tuned LLMs to predict the truth values of state predicates and the validity of actions at intermediate steps of plans. We also quantify world model recovery from the perspective of LLMs as **generative models** (2) by adopting a metric that classifies valid actions and invalid actions using generated token probabilities. To develop our understanding of which aspects of the fine-tuning process impact world model recovery, we fine-tune LLMs using three different data distributions and also explore fine-tuning with a chain-of-thought-style technique where intermediate states are generated between actions. Our experiments, which span two classical planning environments, show that

1. Supervised fine-tuning directly on sequences of valid actions is sufficient for an LLM to learn internal representations that linearly encode action validity *and* the truth values for some (but not all) state predicates.

2. LLMs that have difficulty with using predicted probabilities to distinguish between valid and invalid actions, may still have high quality internal representations capable of making this distinction.

3. Fine-tuning on planning data with good state space coverage (e.g., generated by random walks) generally achieves superior in-distribution and out-of-distribution world model recovery.

Our results are discussed in further detail in Section 6. The rest of this paper is organized as follows. Section 2 provides technical background on classical planning. We describe our SFT setup in Section 3 and the evaluation task and metrics in Section 4. Then, Section 5 introduces our two planning environments and presents planning performance results. Our interpretability results are provided in Section 6. We review related literature in Section 7 and then conclude in Section 8.

## 2 Goal-directed deterministic classical planning

In this section, we introduce the goal-directed deterministic classical planning problem. An example of this type of planning problem is Blocksworld. In a Blocksworld instance, there is a goal—a particular stack of blocks to create—and an initial state—the current location of each block on a flat surface. A Blocksworld plan is a sequence of actions that assembles the target block stack by manipulating blocks one at a time. An action is valid in Blockworld when all of its logical preconditions are satisfied. For example, we can only pick up a block if we aren't already holding another one and if that block has no other block on top of it. Valid Blocksworld plans create the desired block stack by only taking valid actions.

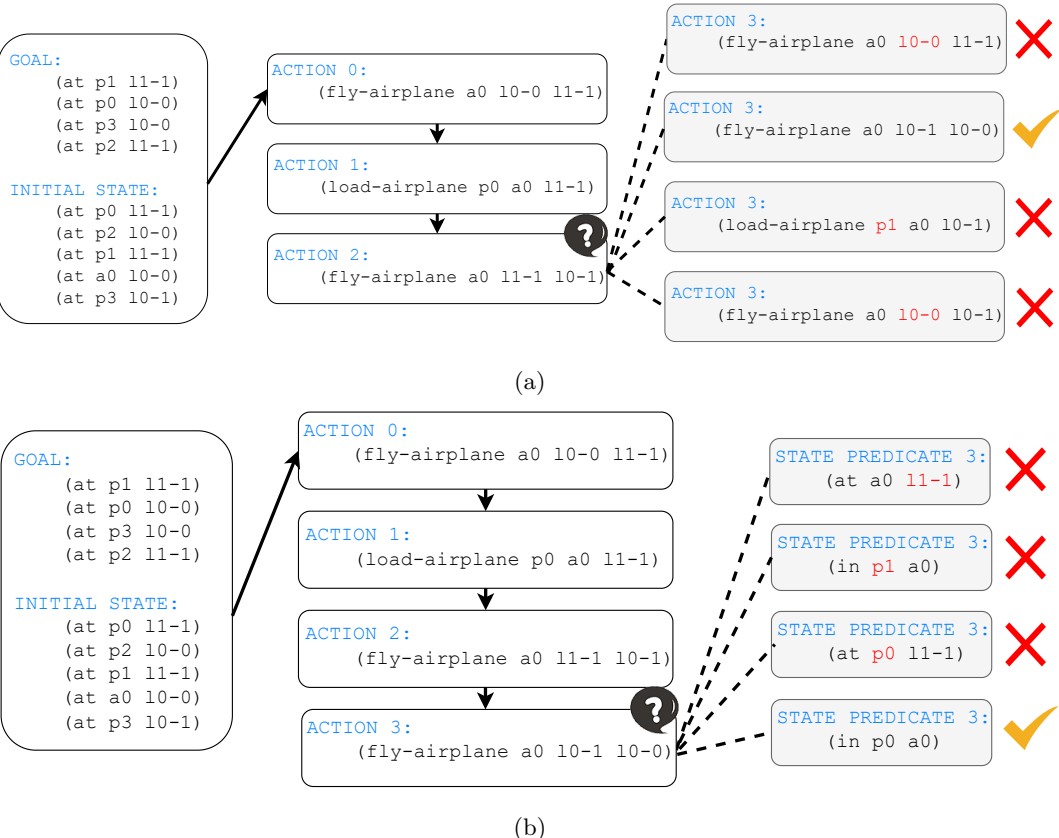

Figure 1: Overview of our interpretability experiments examining world model recovery in supervised fine-tuned LLM planners. We examine both internal representations and generative capabilities to assess how well models capture environment dynamics. **(a)** We study whether planning LLMs proficient at planning can also classify actions as valid or invalid. **(b)** We also investigate whether the internal representations of planning LLMs linearly encode truth values of state predicates. We find that: 1) supervised fine-tuning on valid action sequences enables linear encoding of action validity and certain state predicates; 2) models may learn internal representations that distinguish valid from invalid actions even when output probabilities fail to do so; and 3) broader state space coverage during fine-tuning (e.g., random walk data) improves recovery of the underlying world model.

More formally, each planning environment is described by a planning domain, a set of initial states, and a set of goal states. Planning domains are specified by: a) the states and actions, and b) all state constraints that determine when a certain action is considered valid, and c) a description of all state transitions that result from taking an action. Each planning instances is described by one initial state paired with one goal state. In this work, it is assumed that a PDDL file that provides these details for each planning domain is available. We use STRIPS-like representations of planning problems based on propositional logic, where the domain consists of objects, predicates, and action sets. States are collections of statements that are either true or false. The object set contains all of the objects in the planning environment (e.g., blocks, arm, a table). Predicates are propositional statements about objects in the environment (`on ?x, ?y`), i.e., they are functions of objects. We apply predicates to a specific set of objects to create a statement that is either true or false, e.g., (`on block1, block2`)—either block1 is on block2 and thus the truth value is true, or it is not and the value is false. A unique setting of truth values for all predicates describes a single state in the planning environment. The set of all possible actions that can be taken in the environment is called the action set. Each action has its own set of logical preconditions—predicates that must be true for the action to be valid in a state—and action effects—predicates that become false or true after the action is taken. A

plan that optimally solves a planning instance is the shortest sequence of valid actions from the initial state to the goal state.

# 3 Supervised fine-tuning for end-to-end plan generation

We study *end-to-end plan generation* with LLMs, a setting where LLMs are tasked with generating the entire sequence of actions to arrive at a goal state starting from an initial state, without relying on sophisticated interactions with external solvers. This is a challenging reasoning problem for LLMs, as demonstrated by benchmarks in both the classical setting (Valmeekam et al., 2023; 2024) and the natural language planning setting (Zheng et al., 2024). Other work seeking to improve planning performance has explored alternative paradigms such as LLM-Modulo (Kambhampati et al., 2024), translation of natural language to PDDL (Zuo et al., 2024), and hybrid integrations of search algorithms such as Monte Carlo Tree Search with LLMs (Schultz et al., 2024; Chen et al., 2024). We focus on plan generation because our goal is to study LLM world model recovery, not to improve planning performance, and we argue plan generation is the simplest and most fundamental application of LLMs to planning for our investigation.

Inspired by recent successes in fine-tuning for reasoning, we use SFT to teach `gemma2-9b-instruct` how to do end-to-end classical plan generation. A fine-tuning dataset $D$ of $i = 1, \ldots, |D|$ consists of plans, which are tuples of text $(g^i, s^i, a_1^i, \ldots, a_n^i)$, where $g^i$ is a goal state, $s^i$ is an initial state, and $a_1^i, \ldots, a_n^i$ are a sequence of valid actions that make up an entire plan. Each sequence $(g^i, s^i, a_1^i, \ldots, a_n^i)$ is encoded as text using STRIPS-like notation (see Figure 1 for an example). A state $s^i$ is a list of the predicates that are true at the current plan step.

**Choice of base LLM**: Our fine-tuning approach loosely follows Pallagani et al. (2022), thus, we call our fine-tuned LLMs "Plansformers". We use `gemma2-9b-instruct` as the base LLM, which has been pretrained on vast amounts of code and math (highlighted in Pallagani et al. (2022); Huang et al. (2024) as critical for plan generation) and general text, and fine-tuned with instruction-following data. After fine-tuning, our Plansformers achieve near perfect in-distribution plan generation performance (Section 6). We also attempted training a GPT-2 size transformer from scratch (Radford et al., 2018; 2019) and fine-tuning Llama 3 (Dubey et al., 2024) on our training datasets, but these models struggled to approach a reasonable level of performance (similar observations with Llama 3 in Li et al. (2024))[1]. We encode examples for SFT using `gemma2-9b-instruct`'s instruction template and tokenize each input sequence using the default tokenizer. The next-token prediction fine-tuning objective is computed over the entire sequence.

## 3.1 Training data distributions

Each SFT training example for end-to-end plan generation contains a plan. The mechanism used to generate these training plans is a critical design choice in our empirical study, because LLMs are known to be sensitive to the training distribution used when fine-tuning to enhance reasoning (Zhang et al., 2022). The most common approach is to use optimal or near-optimal plans generated using a traditional plan solver (Pallagani et al., 2022). However, recent studies on LLM world model recovery in board games found that training on optimal actions is *not* conducive to learning good world model representations. Rather, *random valid actions* appear to achieve the best recovery (Li et al., 2022; Vafa et al., 2024). Evidence suggests this is because optimal action sequences encourage learning representations that only help predict *strategically good actions*, rather than *all valid actions* (e.g., all next legal moves from the current board game state) (Li et al., 2022). Random plans can be generated by sampling a random walk starting from any state. In our study, we fine-tune LLMs on both optimal plan data and random walk plan data. We additionally fine-tune LLMs on enhanced random walk plans modified to increase state space coverage (described in detail next). We visually compare these three data generating processes in Figure 2.

**1) Optimal plans**: We obtain these planning problem instances by randomly generating their PDDL files using the `pddlgenerators` library (Seipp et al., 2022). Then, we use the heuristic solver in the popular Fast

---

[1] Although not used in our work, `Qwen2-7B-Instruct` (Team, 2024) is used in the recent fine-tuning study by Huang et al. (2024).

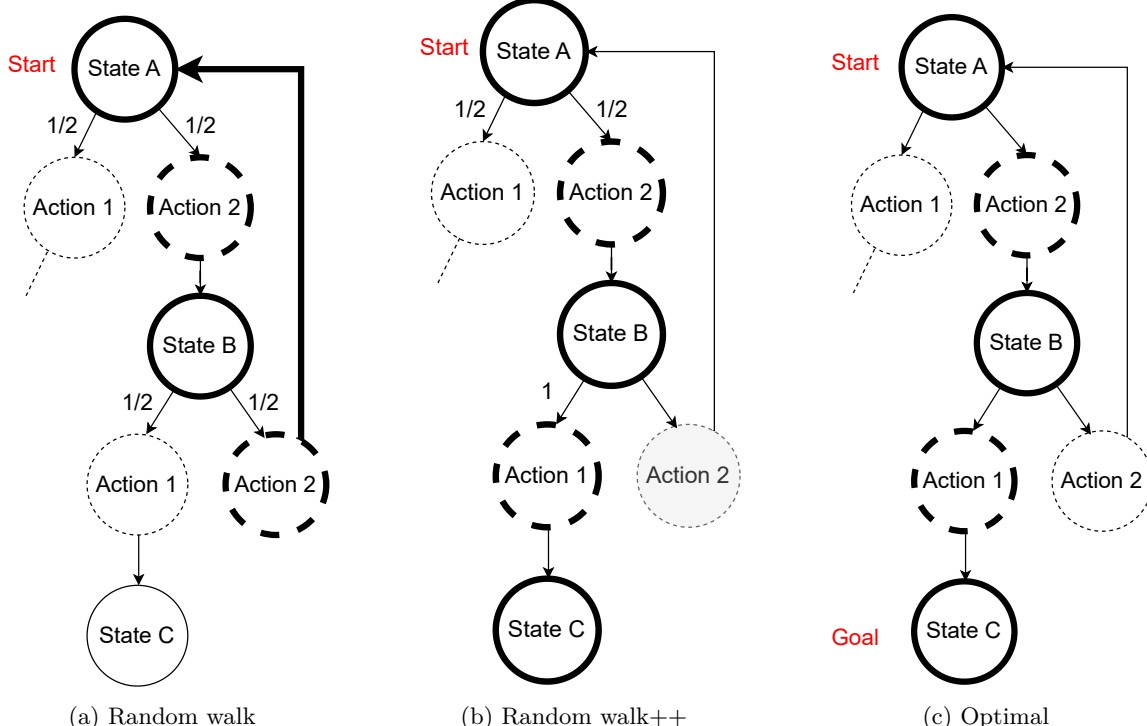

(a) Random walk  (b) Random walk++  (c) Optimal

Figure 2: A toy example that illustrates how plans for supervised fine-tuning are generated by the three different data generating processes. **a) Random walk**: In each state, there are two possible actions, and a uniform random walk samples one with 50% probability assigned to each. Action 2 in State B transitions deterministically back to State A, creating a cycle. A random walk plan starting from State A (bold) may never reach State C before a maximum number of allowed steps: It may start with Action 2 in State A, then Action 2 in State B, then Action 2 in State A, then Action 2 in State B, etc. **b) Random Walk++**: To create cycle-free random walks which explore the state space better, we invalidate Action 2 in State B, because it leads back to State A, a state which has already been traversed. Thus, Action 1 in State B is selected with 100% probability. In this example, this leads to generating the plan in bold, which is the same as the optimal plan (c). **c) Optimal**: Given an initial state (State A) and goal state (State C), we use a heuristic-search-based planner to obtain plans. The plan in bold is used for training, which we spell out here: Action 2 in State A, then Action 1 in State B, which leads to State C.

Downward planner[2] (Pallagani et al., 2022; Li et al., 2024; Huang et al., 2024) as a source of ground truth plans. We refer to these plans as "optimal plans" hereafter.

**2) Random Walk plans**: Random walks are generated by starting at one random initial state $s_0$ and uniformly at random sampling a valid action $a$ up to a maximum number $n \sim \texttt{Unif}(2, 5)$ of steps. Once the maximum steps have been reached, the final state $s_n$ reached by the random walk becomes the new goal state $g$ for that plan. We then set the initial state of the next random walk to be the final state from the previous random walk, i.e., $s_0' = s_n$. We repeat this process until we obtain a desired number of random walk plans.

**3) Random Walk++ plans**: We observed that vanilla random walks in the considered planning domains tend to contain a high number of *state cycles*. A state cycle occurs when, after taking a valid action and transitioning from state $a$ to state $b$, the next valid action causes a transition from state $b$ back to $a$. A high prevalence of state cycles in random walk data leads to poor state space coverage—the training data fails to explore regions of the state space far from the initial states. Concretely, we calculated that for random walks generated with maximum steps $n \sim \texttt{Unif}(2, 11)$ the median plan has a ratio of unique states visited

---

[2]www.fast-downward.org/

to plan length of 80%, and the minimum observed ratio across all plans is 40%. We can remove state cycles by keeping track of the states that have been visited so far in a random walk and invalidating valid actions that would transition the random walk to a previously visited state. These plans, like optimal plans, have no cycles by design. As with (2) Random Walk plans, after a maximum number $n$ of steps, the final reached state $s_n$ is set to be the new goal state $g$, and the random walk continues from an initial state $s'_0 = s_n$.

## 3.2 State chain-of-thought (State-CoT)

Recent work has hypothesized that training LLMs to directly predict the effects of actions can reduce hallucinations during planning (e.g., invalid actions) (Schultz et al., 2024) and improve decision making for action selection (Huang et al., 2024) by enhancing groundedness and coherence. In the context of our investigation, it is natural to ask whether fine-tuning the LLM on sequences consisting of actions interleaved with *next states* $(g^i, s_0^i, a_1^i, s_1^i, a_2^i, s_2^i, \ldots, a_N^i)$ enhances world model recovery. Augmenting sequences with fully observed intermediate states closely resembles chain-of-thought (CoT) prompting strategies for planning such as Reasoning via Planning (RAP) (Hao et al., 2023), which helps LLMs display stronger reasoning abilities. While our augmentation approach is most similar to Huang et al. (2024), instead of interleaving the actions with the entire state, they interleave the *change* to the previous state. While this is to mitigate severely increasing sequence length when the number of tokens required to encode each state is high, which can lead to out-of-memory issues, we did not need this with our setup (see Figure 1 for a visual). In our experiments, we refer to models trained with state chain-of-thought augmented sequences with (-State-CoT).

## 4 Evaluating world model recovery

Our study aims to provide a holistic view on world model recovery in Plansformer LLMs by using two distinct yet complementary strategies for evaluating world model recovery: a) linear probing to assess internal representations, and b) action validity classification based on generated token probabilities. We conduct this investigation on multiple Plansformer LLM variants that differ in fine-tuning data distribution and plan augmentation strategy (State-CoT). Thus, our goals are twofold: to characterize, in an absolute sense, the extent of world model recovery in Plansformer LLMs in a base condition (LLMs fine-tuned on random walk plans without plan augmentation), and to compare world model recovery across the conditions. Linear probes themselves have limited capacity to fit complex functions; thus, they help quantify whether an LLM has learned to organize its internal representation of state predicates and action validity in a simple (i.e., linear) way. By contrast, generated token probabilities provide an extrinsic perspective on the recovery of the world model implicitly learned by the LLM (Vafa et al., 2024).

We do not believe it is obvious that we should observe strong world model recovery in Plansformers. First, while the LLMs are only fine-tuned on *valid* plans, our action validity classification metric involves prompting the LLMs with sequences containing invalid actions at test time. Second, in LLMs trained *without State-CoT* and hence only on sequences of valid actions, it is not obvious that they will learn to internally represent and track the truth values of state predicates at intermediate steps of generated plans.

### 4.1 Extraction of activations and token log probabilities

The metrics are all computed over data extracted by prompting our fine-tuned LLMs on test instances and greedily decoding a plan. We restrict our interpretability analyses to *valid* plans, i.e., plans where each greedily decoded action is valid, discarding invalid plans. Note that only a small fraction of generated plans are invalid, as shown in Table 1. We describe how we collect that data now (Figure 3).

1. We first evaluate each LLM on all $R$ tests plan; for plan $i \in \{1, \ldots, R\}$, we prompt it with the goal $g^i$ and initial state $s_0^i$ and generate a complete plan $p^i := a_1^i, \ldots, a_N^i$ of length $n$.

2. After the LLM generates the plan, we randomly select an intermediate step $j \sim \text{Unif}(0, N-1)$ of that plan.

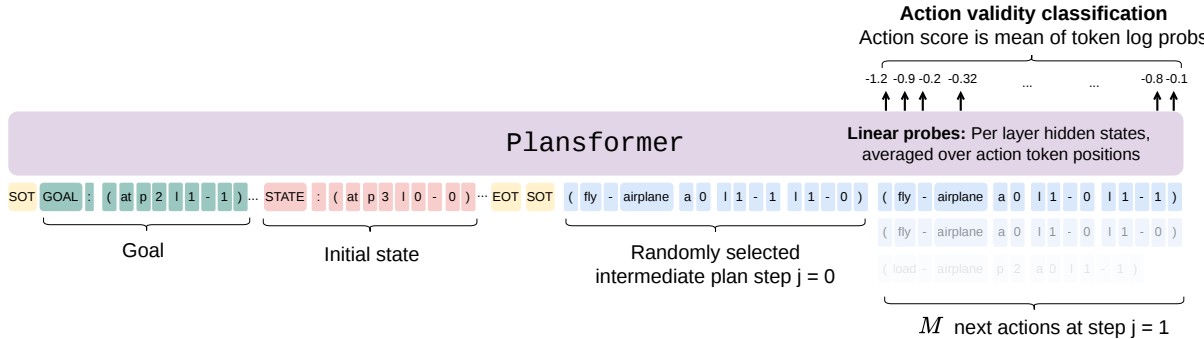

Figure 3: To compute our world model recovery metrics, we extract activations and token log probabilities from Plansformer LLMs prompted on held-out test problem instances. From each LLM-generated plan with $N$ actions for test instance $i$, we create $M$ partial plans consisting of a goal, initial state, the generated actions between steps $0, \ldots, j$, $j \sim \texttt{Unif}(0, N - 1)$, and the $M$ possible next actions at step $j + 1$. We extract $M$ sets of hidden states and average the log probabilities at the token positions corresponding to the appended actions, including '(' and ')' tokens. Here, we visualize an example where $j = 0$.

3. At that intermediate step, we enumerate all possible valid and invalid next actions (suppose there are $M^i$ of them) and create $M^i$ partially completed plans by concatenating the partial plan $p_j^i$ with each possible next action $a_{j+1}^{i,m}$, $m = 1, \ldots, M^i$.

4. Then, we prompt the LLM again, this time with $M^i$ partial plans, $p_j^i a_{j+1}^{i,m}$.

5. At all token positions corresponding to action $a_{j+1}^{i,m}$, we extract both a) hidden states for training linear probes, and b) token log probabilities computed as the log softmax of the logits.

## 4.2 Linear representation probes

Using the activations extracted following the recipe detailed in Section 4.1, we train two types of linear probes: action validity linear probes and state predicate linear probes.

**Action validity linear probes**: We train layer-wise logistic regression probes to predict action validity from activations. With the data from Section 4.1, we create a probe training dataset of size $L \times R \times M$ where $L$ is the number of LLM layers (excluding embedding and unembedding layers) and $M = \max_{i=1,\ldots,R} M^i$. The activations correspond to all action tokens $a_{j+1}^{i,m}$ between the '(' and ')' tokens (inclusive). Each entry in the dataset is an activation vector averaged over action tokens paired with a corresponding binary action validity label. The input to each logistic regression probe (Belinkov, 2022) are the activation vectors averaged over action tokens from layer $l \in L$ of the LLM prompted with partial plan $p_j^i a_{j+1}^{i,m}$, and the output of each probe is the probability that $a_{j+1}^{i,m}$ is valid, which we compare against the binary labels.

**State predicate linear probes**: In classical planning, each state is defined as a unique setting of (a potentially large number of) state predicate truth values. Let $|S|$ be the number of such predicates. To examine whether our LLMs internally represent the state while planning, we train multiple logistic regression probes, one for each state predicate. Each of the $|S|$ probes takes as input the activation vectors averaged across action tokens from LLM layer $l \in L$, where the LLM has been similarly prompted with partial plan $p_j^i a_{j+1}^i$. This probe predicts the probability that the state predicate after applying $a_{j+1}^i$ is true.

**Probe evaluation metric**: Probes are trained by aggregating activations and binary labels into a dataset of size $L \times R$. For both the layer $l$ action validity and state probes, we randomly split the data 80/20 into train/test, and use test F1 as the evaluation metric. We choose F1 because our binary classification setting is both highly imbalanced and the majority label is the negative outcome—invalid actions. Invalid actions often represent over 90% of the labels. Since the minority label is the positive outcome (valid actions), F1 is informative as it balances both positive outcome precision ("out of the identified valid actions, how

many were false positives") and recall ("did we find all of the valid actions"). By contrast, a probe that only predicts the majority label would have a precision of zero and thus an F1 score of zero. Our logistic regression probe uses 10K maximum iterations to ensure convergence, does not fit an intercept, and uses auto-balancing.

### 4.3 Generative metric

We also quantify world model recovery by using generated token probabilities for the appended actions to the partial plan to classify these actions as valid or invalid. In detail, we use data extracted in Section 4.1 to create a dataset of size $R \times M$, where each entry in the dataset is the average of the token log probabilities assigned to the valid and invalid actions $a_{j+1}^{i,m}$ appended to partial plan $p_j^i$. Each action is paired with its binary validity label. As our quantitative classification metric, we compute the fraction of test plans for which *all* valid actions are ranked higher than invalid actions, sorting actions by their average token log probabilities.

## 5   Planning environments

We finetune and evaluate `gemma2-9b-instruct` on two popular planning domains, **Blocksworld** and the more challenging **Logistics**. In the Blocksworld domain, there is a collection of blocks, a flat table, and a set of rules for stacking and unstacking blocks. Our in-distribution data for Blocksworld consists of instances with 3-5 blocks. In any given state, the available actions are: (`put-down ?x`), (`pick-up ?x`), (`stack ?x, ?y`), (`unstack ?x,?y`). The total number of available actions depends on the number of blocks in the planning instance. The Logistics domain resembles a delivery problem, where the goal is to move packages from one set of locations to another. We use a simplified domain which only has cities, airplanes, and packages (with airports and trucks removed to reduce the size of each state when encoded as text), with rules governing the movement of airplanes as well as when packages can be loaded or unloaded. Here, the available actions in any given state are: (`fly-airplane ?a ?loc1 ?loc2`), (`load-airplane ?pkg ?a ?loc`), (`unload-airplane ?pkg ?a ?loc`). We generate in-distribution data with 3-4 packages, thus, the number of available actions per state is also variable.

**Training details**: We generate 6 fine-tuning datasets. For Blocksworld, there are 80,190 random walk plans (1,013 unique goal states), 60,962 random walk++ plans (1,013 unique goal states), and 21,851 optimal plans (500 unique goal state). For the Logistics domain, there are 81K random walk plans (3K unique goal states), 81K random walk++ plans (3K unique goal state), and 16.2K optimal plans (320 unique goal states). To control for the amount of fine-tuning data used across datasets, we randomly down-sample the random walk and random walk++ datasets to have the same number of plans as the optimal datasets. We reserve 166 held out planning problems to use as in-distribution test data. Models with State-CoT have a maximum plan length of 5 to avoid excessively increasing the training sequence length, which caused training instability. All models are trained with default LoRA parameter efficient fine-tuning (Hu et al., 2022) for 4 epochs and an initial learning rate of 3e-4. Models without State-CoT use a batch size of 4 whereas models with State-CoT use a batch size of 2, and all models use gradient accumulation of 16. We repeat each fine-tuning run with three different random seeds and present our results as averages over seeds with standard deviations.

**Planning performance results**: After fine-tuning, LLMs without State-CoT achieve an average goal reach rate and valid plan rate of 99.2% on unseen in-distribution plan instances (Table 1) With State-CoT, the LLMs fine-tuned on random walk and random walk++ datasets saw a performance drop compared to optimal plans. State-CoT introduces a non-trivial increase in the length of training sequences, which can increase the amount of training tokens needed to achieve a certain level of performance. We conducted training runs for State-CoT LLMs with extra random walk and random walk++ plans initially discarded during downsampling, and were able to boost the State-CoT average goal reach rate to 97%. This sets the stage for us to analyze whether LLMs proficient at in-distribution planning have *also* partially, or fully, recovered the underlying planning domain model.

Table 1: **Planning performance**. Plans are generated using greedy decoding, and each result shows mean and std. dev. over three training run random seeds. **Goal reached** is the percent of plans where the last state of the generated plan is the goal state, and **valid** is the percent of plans that have no invalid actions (LLMs are known to occasionally generate plans that reach goals while "hallucinating" intermediate invalid actions (Momennejad et al., 2023)). **Bad state** shows the percent of plans where models trained with State-CoT predict an incorrect state transition. All of our (non-State-CoT) `gemma2-9b-instruct` Plansformer variants achieve nearly perfect planning goal reach rate and valid plan rates on in-distribution test plans.

| Training data | Model type | Goal reached$_\uparrow$ (%) | Valid$_\uparrow$ (%) | Bad state$_\downarrow$ (%) |
|---|---|---|---|---|
| **BlocksWorld** (3-5 blocks, $R = 166$ plans) | | | | |
| Random walk | Plansformer | $97_{\pm 3}$ | $97_{\pm 3}$ | - |
| Random walk | Plansformer+State-CoT | $75_{\pm 26}$ | $82_{\pm 19}$ | $15_{\pm 19}$ |
| Random walk++ | Plansformer | $99_{\pm 1}$ | $99_{\pm 1}$ | - |
| Random walk++ | Plansformer+State-CoT | $98_{\pm 2}$ | $98_{\pm 2}$ | $0_{\pm 1}$ |
| Optimal | Plansformer | $100_{\pm 1}$ | $100_{\pm 1}$ | - |
| Optimal | Plansformer+State-CoT | $92_{\pm 11}$ | $95_{\pm 8}$ | $3_{\pm 4}$ |
| **Logistics** (3-4 packages, 1 airplane, 2 cities, 2 locations, $R = 166$ plans) | | | | |
| Random walk | Plansformer | $100_{\pm 0}$ | $100_{\pm 0}$ | - |
| Random walk | Plansformer+State-CoT | $5_{\pm 1}$ | $97_{\pm 3}$ | $3_{\pm 3}$ |
| Random walk++ | Plansformer | $100_{\pm 0}$ | $100_{\pm 1}$ | - |
| Random walk++ | Plansformer+State-CoT | $2_{\pm 1}$ | $63_{\pm 33}$ | $29_{\pm 0}$ |
| Optimal | Plansformer | $99_{\pm 1}$ | $99_{\pm 1}$ | - |
| Optimal | Plansformer+State-CoT | $98_{\pm 2}$ | $100_{\pm 1}$ | $1_{\pm 1}$ |

## 6 Main results

In the previous section, we demonstrated that most of our supervised fine-tuned LLM planners achieve near optimal planning performance on held-out, in-distribution test instances. Now, we conduct interpretability studies to explore whether these same LLMs have also recovered a planning world model.

### 6.1 Plansformers learn linear representations of action validity

The probe test F1 scores on held-out, in-distribution test plans, visualized in Figure 4, indicate that Plansformers learn to encode the validity of actions linearly. We also see that across models, linear probe scores tend to plateau around the middle layers, indicating that intermediate and upper layers better encode action validity than lower layers. No significant difference in action validity probe performance when using State-CoT is observed. Across the fine-tuning setups we explored, random walk++ training data—the exploration-enhanced random walk plans—achieves the best reliability across training seeds (e.g., Figure 4d).

### 6.2 Plansformers learn to linearly encode the truth values of certain state predicates

Plansformers—LLMs fine-tuned on sequences of valid actions—learn internal representations that linearly encode the validity of the next action. Do they also learn linear representations of state predicates at intermediate plan steps, without direct supervision? Figure 5 visualizes state probe F1 scores for Plansformers *without State-CoT*, fine-tuned on random walk data. Given the large number of predicates (e.g., in Blocksworld with 3-5 blocks there are at most 36 state predicates), for presentation clarity we aggregate the predicate F1 scores across objects. However, we do not train or evaluate any state probes that, for a given plan, correspond to an irrelevant predicate. For example, for a BlocksWorld problem with only 4 blocks, we do not probe for grounded state predicates that have `block5`. In Blocksworld (Figure 5a), the F1 scores suggest that the internal representations linearly encode most state predicates besides `(on ?obj1, ?obj2)`. A similar trend appears in Logistics, where the predicate `(at ?pkg, ?loc)`, which has two arguments that varies across objects, has the lowest F1 scores. Since we only have a single airplane `a0` in our Logistics en-

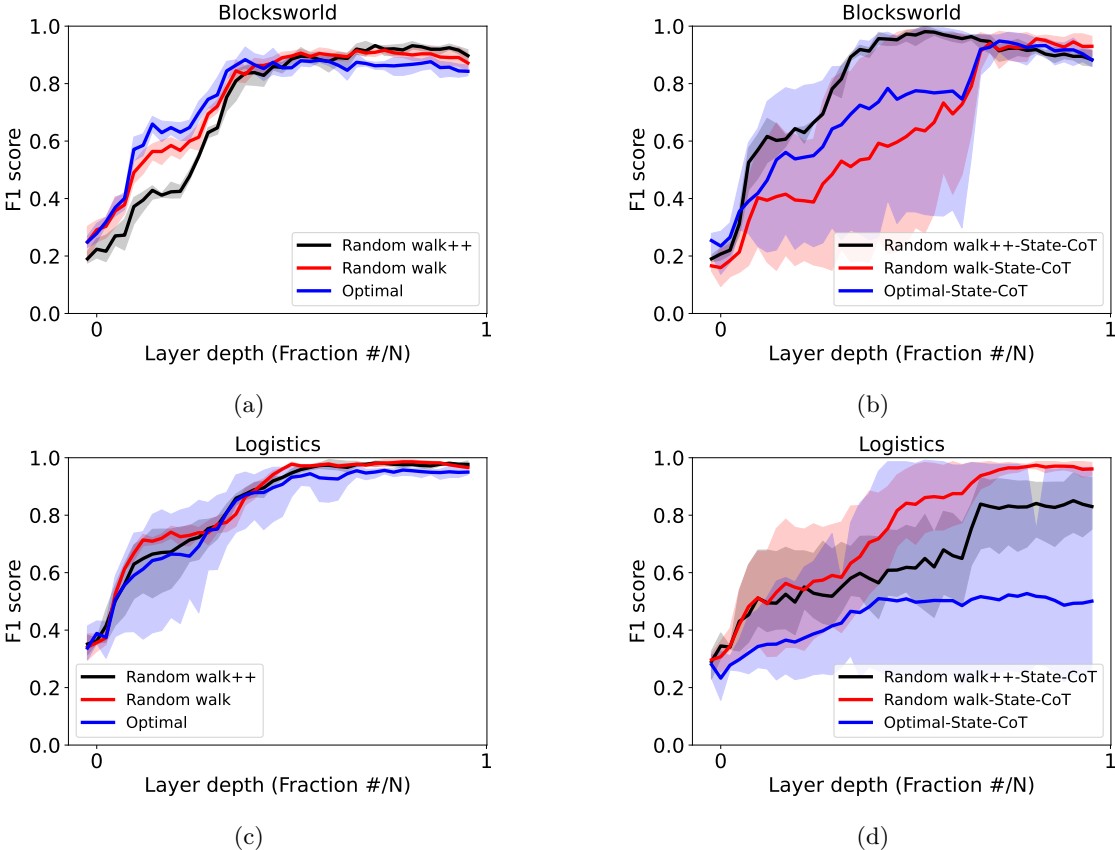

Figure 4: **Action validity linear probe results**. We train linear probes on internal representations of Plansformers to predict the validity of next actions from an intermediate plan step. Each line plot shows mean and 95% confidence intervals across three training seeds. (a) and (c) are models fine-tuned without State-CoT, whereas (b) and (d) use State-CoT. Across all fine-tuning configurations, probe F1 scores reach ∼1.0 by the intermediate and highest layers.

vironment, the predicates (`at ?airplane, ?loc`) and (`in ?pkg, ?airplane`) always have one argument fixed. Overall, state predicate probe scores are high for certain predicates and low for others, and we observe a trend where simpler (i.e., fewer arguments) state predicates have the higher scores. This trend may be explained by differences in the balance of positive and negative labels during probe training; for example, we observe that the state predicate (`arm-empty`) has a roughly even label distribution and attains by far the highest probe scores. Our results, which show uneven probe accuracy across state predicates, suggest only partial world model recovery in our Plansformers.

## 6.3   Training on random walks enhances action validity classification

Whereas in Sections 6.1 and 6.2 we presented the results of our linear probe metrics for evaluating world model recovery in Plansformers, here we examine our generative action validity classification results. Examining the action token log probabilities shows that fine-tuning with uniform (unmodified) random walk plans leads to relatively better classification results by a significant margin (Table 2, Figure 6). On average, LLMs fine-tuned on random walk plans score all valid actions as more probable than the invalid actions ∼81% of the time, compared to ∼29% for training with random walks with enhanced exploration (random walk++) and just ∼9% for optimal plans. To test whether the difference in action classification performance between random walk and random walk++ data is statistically significant, we ran a Mann-Whitney U-test. Three out of four cases had statistically significant ($p \leq 0.05$) differences—only the Logistics environment with State-CoT had $p > 0.05$ ($p = 0.055$).

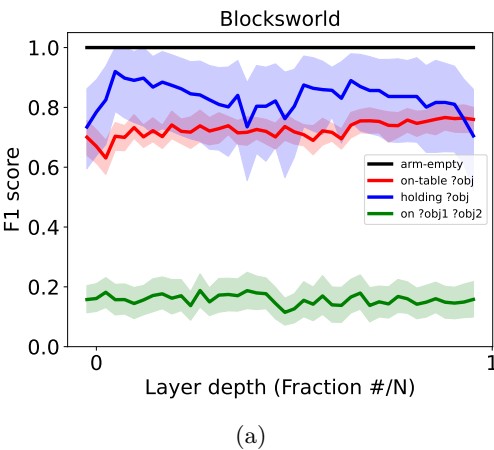 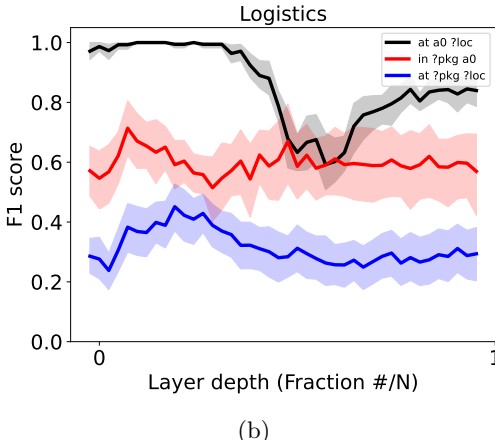

(a)                                               (b)

Figure 5: **State predicate linear probe results**. We predict the truth values of state predicates at intermediate plan steps using linear probes trained on Plansformer activations without State-CoT. The Plansformers are fine-tuned on uniform random walk data. Probe scores for individual predicates are averaged over objects for presentation clarity. The scores vary widely by predicate, e.g., in Blocksworld (5a) the `(arm-empty)` predicate is predicted perfectly while (`on ?obj1, ?obj2`) is not predicted with good accuracy.

Table 2: **Action classification results**. Percent of test plans where, when prompted at a randomly selected plan step, all subsequent valid actions are ranked higher than invalid actions. Actions are scored by averaging over the action token log probabilities. Highlighted cells are statistically significant ($p \leq 0.05$) results from a Mann-Whitney $U$-test comparing the distributions of action ranks between Random walk and Random walk++ training data.

| Model type | Random walk | Random walk++ | Optimal |
|---|---|---|---|
| **Blocksworld** (3-5 blocks) | | | |
| Plansformer | $80_{\pm 2}$ | $36_{\pm 4}$ | $9_{\pm 1}$ |
| Plansformer+State-CoT | $83_{\pm 2}$ | $20_{\pm 1}$ | $19_{\pm 4}$ |
| **Logistics** (3-4 packages, 1 airplane, 2 cities, 2 locations) | | | |
| Plansformer | $82_{\pm 1}$ | $33_{\pm 2}$ | $4_{\pm 1}$ |
| Plansformer+State-CoT | $79_{\pm 13}$ | $27_{\pm 10}$ | $4_{\pm 0}$ |

Kernel density estimates of the mean valid action token log probabilities, aggregated across planning environments, are visualized in Figure 6. Fine-tuning with optimal plans fails to encourage the LLM to generate valid actions with mean token log probabilities higher than invalid actions (Figure 6). This conflicts with the action validity probing results (Figure 4), which show that these LLMs do possess internal representations that linearly encode action validity. Our results here corroborate with findings from other LLM interpretability studies that reveal that the internals of LLMs "know more than they say", e.g., about the correctness of their predictions (Orgad et al., 2024; Liu et al., 2025).

The action validity classification results for optimal data are in line with observations made in Toshniwal et al. (2022) about legal chess move prediction performance. Using only human chess games as training data for a transformer is limiting because it only has "meaningful" legal moves rather than examples of all legal moves. It is possible that fine-tuning with random walk plans achieves the best relative world model recovery under this metric because it pushes the LLM to assign high probability to *any* valid action. The gap between random walk and random walk++ data here may be explained by the fact that we remove cycles in training plans by disallowing certain valid actions, as we show in Figure 2. While we expect it is possible to improve our fine-tuning approach so that this gap goes away, we leave exploring this to future work.

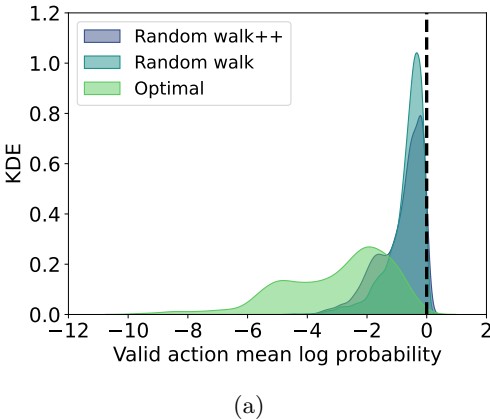 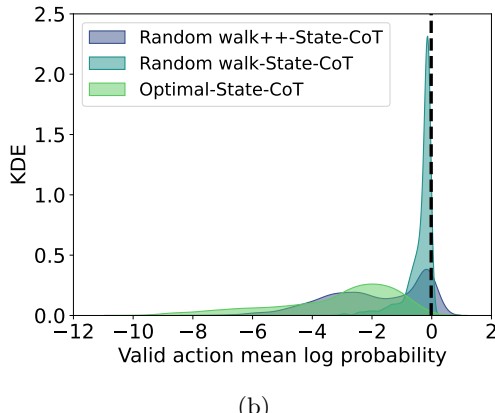

(a)                 (b)

Figure 6: **Kernel density estimates for valid actions**. Each data point in the density estimate is the average over a valid action's token log probabilities. Note that we only include alternative actions to the greedily decoded action in these density estimates; models always assigned $\sim 0$ log probability to the greedily decoded action tokens. We combined Blocksworld and Logistics results to create each density estimate. The vertical dashed line shows the gold standard—assigning an average log probability of 0 to each token of a valid action. Density to the right of 0 is an artifact from the KDE fit. Plansformers fine-tuned with optimal plans cannot distinguish valid and invalid actions, assigning negligible probability to both valid actions (aside from the greedily decoded action) and invalid actions. Plansformers fine-tuned on uniform random walks excel under this metric.

Table 3: **Out-of-distribution generalization planning performance**. We evaluate LLMs fine-tuned on Blocksworld problems with 3-5 blocks on OOD instances with 6-10 blocks, from the PlanBench (Valmeekam et al., 2024) benchmark. We modify their instances to use STRIPS-like text formatting. The LLMs fine-tuned on random walks with enhanced state exploration are more robust to distribution shifts.

| Training data | Model type | Goal reached$_\uparrow$(%) | | | | | Valid $_\uparrow$(%) | | | | |
|---|---|---|---|---|---|---|---|---|---|---|---|
| Number of blocks | | 6 | 7 | 8 | 9 | 10 | 6 | 7 | 8 | 9 | 10 |
| Random walk | Plansformer | $0_{\pm 0}$ | $0_{\pm 0}$ | $0_{\pm 0}$ | $0_{\pm 0}$ | $0_{\pm 0}$ | $72_{\pm 11}$ | $60_{\pm 19}$ | $55_{\pm 17}$ | $51_{\pm 21}$ | $58_{\pm 14}$ |
| Random walk++ | Plansformer | $53_{\pm 5}$ | $28_{\pm 4}$ | $6_{\pm 6}$ | $2_{\pm 3}$ | $0_{\pm 0}$ | $53_{\pm 4}$ | $30_{\pm 2}$ | $14_{\pm 6}$ | $16_{\pm 5}$ | $4_{\pm 1}$ |
| Optimal | Plansformer | $59_{\pm 27}$ | $31_{\pm 17}$ | $14_{\pm 14}$ | $6_{\pm 5}$ | $6_{\pm 6}$ | $64_{\pm 26}$ | $40_{\pm 20}$ | $23_{\pm 20}$ | $16_{\pm 8}$ | $10_{\pm 8}$ |

## 6.4 Does better world model recovery imply better OOD generalization?

Up to now, our analysis has been on held-out *in-distribution* plans. Do the LLMs that demonstrate better in-distribution world model recovery, as judged by our linear probing and action classification metrics, also generalize to *out-of-distribution* (OOD) plans? Although we do not go so far as to explicitly test for a causal link between the linear internal representations from Section 6.1-6.2 and the LLM's predictions, strong OOD performance would provide some evidence that the LLM might be using a world model to plan, as world models encode environment transition dynamics which are unaffected by variations such as an increase in the number of objects.

For this experiment, we use BlocksWorld problem instances from the PlanBench benchmark (Valmeekam et al., 2024) with 6-10 blocks (a total of 261 instances), up to five more than the maximum number of blocks seen during fine-tuning. We evaluate the LLMs fine-tuned on BlocksWorld without State-CoT on this OOD data. We did not evaluate LLMs fine-tuned with State-CoT because their performance on this OOD task is confounded by the challenge of *length generalization*. The number of state tokens that are generated for each plan step inflates the size of the generated plans dramatically; specifically, OOD plans have roughly 2.5× more tokens than in-distribution plans.

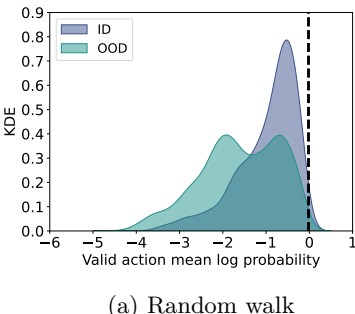 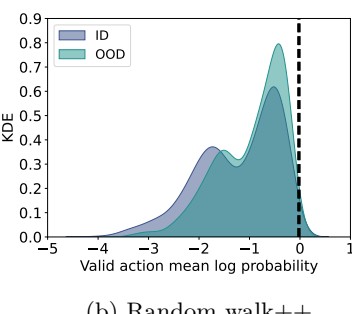 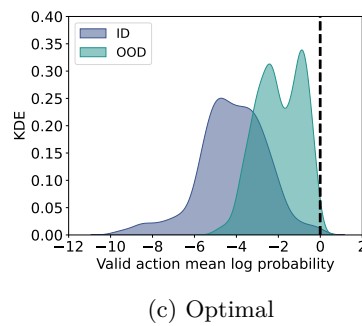

(a) Random walk        (b) Random walk++        (c) Optimal

Figure 7: **ID vs. OOD kernel density estimates for valid actions**: Out-of-distribution Blocksworld planning instances with 6-10 blocks, compared across training data used for fine-tuning (a-c). We overlay the ID and OOD KDE plots for valid actions. Plansformers fine-tuned on random walk data (7a) show a collapse (leftward shift) in their ability to distinguish between valid and invalid actions by token probabilities. Fine-tuning on random walk++ data (7b, no shift) and optimal plan data (7c, moderate rightward shift) display a robustness to distribution shift. Note that the model fine-tuned on optimal data (7c) has poor performance under this metric in both ID and OOD settings.

Table 3 shows the planning performance. Plansformers fine-tuned with random walk++ and optimal data see similar drops in performance when introducing a sixth block, followed by a steady decline as the number of unseen blocks increases, to near zero goal reach rate with ten blocks. LLMs fine-tuned on uniform random walks fail to reach any goals (0% OOD goal reach rate). The KDE fit to the valid action mean token log probabilities for random walk++ data is mostly unchanged on OOD data (Figure 7b), while the KDE fit to random walk data shows a strong leftwards shift (Figure 7a) and the results for optimal data show a moderate rightwards shift. **In our setup, Plansformers fine-tuned on uniform random walks lose the ability to distinguish valid and invalid actions by token log probability on the OOD split.** It is plausible this is because the distribution shift from the random walk training data to the PlanBench OOD test data is largest; our intervention on the data generation process (random walk++) has a significant effect on OOD performance. However, all models maintain good linear action validity probe scores on

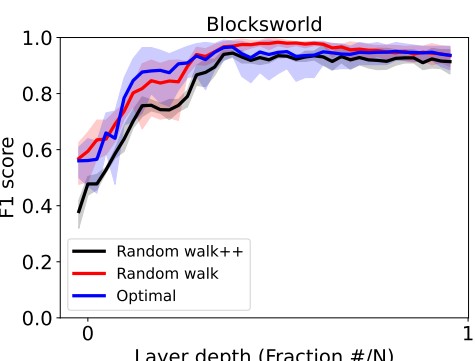

Figure 8: **Out-of-distribution action validity linear probes**. Testing on Blocksworld instances with 6-10 blocks. The internal representation quality is maintained on OOD plans.

OOD data (Figure 8), *even the Plansformers trained on random walk data*, highlighting again a mismatch between internal representation quality and generative capabilities.

## 7 Related Work

Various recent studies, which we review here, have also explored whether transformers and LLMs learn representations that encode meaningful aspects of the world in other contexts (Li et al., 2021; Gurnee & Tegmark, 2023). Advances in the capabilities of LLMs has spurred interest in this question, since its implications are significant (Yildirim & Paul, 2024). Related empirical studies have examined transformers trained on sequential decision making tasks but have been limited to using relatively small transformers trained from scratch on board games (Toshniwal et al., 2022; Li et al., 2022; Nanda et al., 2023; Yuan & Søgaard, 2025; Yun et al., 2023; Jenner et al., 2024; Karvonen, 2024) and navigation-style problems (Ivanitskiy et al., 2023; Dedieu et al., 2024; Vafa et al., 2024; Brinkmann et al., 2024). It is not clear whether the observations from those studies extend to the planning problems we are interested in, whose complexity necessitate the use of pretrained language models (Li et al., 2024). An interpretability study by Men et al. (2024) asks whether

fine-tuned `llama2-7b-chat` and `vicuna-7b` models have and use a look-ahead planning mechanism, but simplify the planning task significantly by converting it into a fill-in-the-blank classification problem.

One-step action validity has been used to evaluate world model recovery previously, e.g., in chess by looking at the top-$k$ predicted next moves of chess-playing transformers, where $k$ is equal to the true number of all valid moves for a chess piece at a particular board state (Toshniwal et al., 2022; Schultz et al., 2024). Other, more complex tasks for inspecting world model recovery such as multi-move lookahead (Jenner et al., 2024; Men et al., 2024) and world model compression and distinction (Vafa et al., 2024) have also been proposed in the literature. In our fully observable and deterministic classical planning problems, the PDDL model contains a complete description of the domain's transition dynamics. In fact, it is known that PDDL models can be equivalently represented as deterministic finite automata (DFA) (Toropila & Barták, 2010), and Vafa et al. (2024) introduced the idea that a generative model has implicitly recovered a world model described by a DFA when the model generates a sequence with positive probability if and only if it is also valid in the DFA. Applying this metric in our classical planning framework would be a natural extension of the generative one-step action classification metric we employ, which we leave for follow-on investigation.

The most closely related work to ours is Hirsch et al. (2024). They perform experiments investigating planning world model recovery in pretrained language models, finding that such models have a poor ability to both 1) predict the effects of actions and 2) predict the validity of actions. Our work differs in various aspects of the experimental setup and thereby also in our findings. First, they conduct their study on pretrained models without fine-tuning, and as our work shows, after fine-tuning with carefully curated training data we observe (partial) world model recovery in LLMs. Second, they only examine model predictions; as our work shows, it is important to also analyze internal representations, which may not agree with extrinsic metrics based on model predictions. While Hirsch et al. (2024) and our own work find that planning performance degrades on OOD data, our findings are more optimistic, as we show that our LLMs have internal representations robust to these (small) distribution shifts, and that enhancing exploration in random walk training data improves OOD planning performance.

## 8 Conclusion

Does supervised fine-tuning LLMs on examples of plans (so-called Plansformers) learn to recover the planning world model? Our work investigated this question empirically with the open model `gemma2-9b-instruct` by designing and conducting a series of rigorous experiments across two planning environments and with metrics that look at both internal representation quality and generative properties. Our results are optimistic, showing that we can fine-tune LLMs to be proficient planners *and* to partially recover the planning world model on held-out, in-distribution planning problems. This includes observing a strong ability to linearly predict action validity from internal representations and an ability to predict the truth values of certain *state predicates* at intermediate plan steps, despite only being trained to predict valid *actions*. The story is more complicated for out-of-distribution (OOD) planning problems. OOD experiments show that internal representations maintain their linearly encoding of action validity, but the planning performance and generative world model recovery metric for Plansformers fine-tuned on random walks, the best in-distribution model, collapses. This collapse is somewhat mitigated by an intervention on the random walk data generation process that enhances the state space coverage in the resultant training set (random walk++).

The results of our study should be taken in the context of key limitations. First, we only show the *existence* of internal representations that encode action validity and state predicate truth values, but do not go so far as to show a causal link between these representations and the model's predictions. Ultimately, it is unclear if the representations we found in fine-tuned LLMs are implicated in the LLM's planning process. Second, it is unknown whether our findings generalize to LLMs that have significantly more parameters. Third, there are a large number of experiment design decisions that were made to conduct our study, and while we do discuss and justify many of them, we were not able to control for the effects of all of them. For example, we focused our interpretability analyses on only valid LLM-generated plans, which adds a layer of context to our results that assumes all analyzed plans are "good" plans. An immediate follow-up study could try isolating the last valid plan step within an *invalid* plan to analyze world model recovery under a relaxation of this assumption. Fourth, our investigation focuses on plans encoded with a constrained, STRIPs-based text

template instead of natural language, which may inhibit LLMs from fully making use of their commonsense knowledge learned during pretraining.

To conclude, we believe the findings of our work are optimistic about the viability of fine-tuning to enhance planning in LLMs, and encourage the exploration of advanced strategies such as reinforcement learning fine-tuning (Huang et al., 2024) to further push the OOD performance of end-to-end LLM planners.

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
