# OpenReview forum: "A Close Look At World Model Recovery In Supervised Fine-Tuned LLM Planners"
_TMLR — Under review for TMLR_

### Review · Reviewer_Csk6 · 2026-01-19

**Summary Of Contributions:**

In this work, the authors investigate the extent to which large language model (LLM) are able to model an internal representation of the world (in the form of a world model) after fine-tuning the LLM on PDDL-based planning tasks. Using an open-sourced LLM (Gemma-2-9B model), the authors are able to probe the internal dynamics of the model, giving us key insights on the inner workings of the model.

Strengths:

1. By using a regression based method for probing, and a kernel based method for analyzing the generative belief, the work identifies a key area of bottleneck, where internal knowledge does not translate to output performance.
2. The use of Random Walk ++ is quite innovative and successfully mitigates the state cycle biases demonstrating a deeper understanding of the pitfalls in classical planning.

Weaknesses:

1. The authors have already mentioned this limitation in the conclusion, but one of the biggest limitation is that the lack of a causal link between the internal representations of the model and the output predictions. A good experiment would be to perform some sort of interventional experiments (activation patching, causal tracing) where the authors flip the weight of the valid action to invalid in the hidden layer, and checking if the LLM stops itself from taking the action? If yes, that can be a source of causation and not simply correlation.

2. For any planning task, the number of invalid actions are much higher than the number of valid actions. Thus the model can get a high F1 score by just being more negative in its search for actions. A good experiment would be to see how does the F1 score of the linear probing approach compare to a simple baseline that will always predict 'invalid'? A precision-recall curve will be more informative than a simple F1 score in this imbalanced setting.


Other:

1. The jump from 5 blocks to 6 blocks is a relatively small distribution shift compared to the exponential growth observed in other planning settings.
2. Using only one airplane for the logistics domain seems like a simplified version of the experiment.

**Audience:**

Yes

**Audience Explanation:**

Yes, I believe this work is relevant to the greater TMLR community as it provides an interesting take on the world model representation for large language models. This work sits at the intersection of the planning ability of large language models, machine learning, classical planning and world models.

The findings of this work are empirical in nature. Through a series of experiments, the authors evaluate world model recovery through two types of experiments: internal representations and generative behavior.

I believe these findings would be relevant for the interests of the broader TMLR community.

**Broader Impact Concerns:**

A broader impact statement is not present and not required.

**Claims And Evidence:**

Yes

**Claims Explanation:**

The authors provide a empirical analysis between the LLM's internal representations and external outputs. The evidence of the internal representation is well studied using the linear probing approach across multiple layers. The evidence for the generative gap is also well studied using the kernel density estimates, looking at how different training distributions affect the model's confidence in valid actions. Overall, the test methodology follows a scientific and rigorous protocol to justify the claims mentioned in the paper.

**Requested Changes:**

Below is a list of requested changes (all are derived from the weakness part of the contributions summary above):

1. Causal Interventions: It would be very interesting to have an experiment that tests activation patching or causal tracing to prove that the identified internal representations functionally drive the model's output. This experiment is important to move beyond the perceived correlation and towards causation.

2. Imbalance Baselines: I believe it is important to address the high ratio of invalid-to-valid actions by comparing linear probe performance against a simple baseline that always predicts invalid actions. Providing Precision-Recall curves are crucial to clarify the model's accuracy in identifying rare valid actions.

3. Extended OOD Testing: If possible, it would be nice to have the work tested on larger problem sizes (e.g., 8–10 blocks) to verify if the world model recovery is robust.

4. Logistics Complexity: Again, if possible, it would be nice to evaluate the model in a more complex logistics setting with multiple airplanes. The current single-vehicle setup simplifies the world model recovery task by removing the scenarios that require resource coordination.

---

> ### Author Response · Authors · 2026-04-12
> **Author Response**
>
> Dear Reviewer Csk6,
>
> Thank you for reviewing our manuscript. We have updated our manuscript with changes made in blue. Here we summarize the key changes based on your review. We also attempt to provide more context and justification for why we did not conduct certain requested experiments.
>
> **[Extend OOD Testing]** To strengthen our OOD experiments we have added additional OOD plans with 7-10 blocks. Section 6.4 and Table 3 have been updated with the new results. We saw no change in world model recovery metrics. The majority of the new text added to Section 6.4 reads: *"We did not evaluate LLMs fine-tuned with State-CoT because their performance on this OOD task is confounded by the challenge of \emph{length generalization}. The number of state tokens that are generated for each plan step inflates the size of the generated plans dramatically; specifically, OOD plans have roughly 2.5 more tokens than in-distribution plans."* and *"Table 3 shows the planning performance. Plansformers fine-tuned with random walk++ and optimal data see similar drops in performance when introducing a sixth block, followed by a steady decline as the number of unseen blocks increases, to near zero goal reach rate with ten blocks.".*
>
> **[Imbalance baselines]** We can justify our choice of F1 metric, which is particularly well-suited for highly imbalanced binary classification settings where the minority class is the positive label (valid action). We added this text to Section 4.2: *"We choose F1 because our binary classification setting is both highly imbalanced and the majority label is the negative outcome---invalid actions.
> Invalid actions often represent over 90\% of the labels. Since the minority label is the positive outcome (valid actions), F1 is informative as it balances both positive outcome precision ("out of the identified valid actions, how many were false positives") and recall ("did we find all of the valid actions"). By contrast, a probe that only predicts the majority label would have a precision of zero and thus an F1 score of zero."*
>
> **[Causal interventions]** While we respectfully agree with the reviewer that a causal intervention study would be highly interesting, we intentionally left this for future work as it is *highly non-trivial to accomplish this for classical plan generation with LLMs*. For example, as our results show, Plansformer LLMs learn internal representations of both action validity and state predicate truth. It is plausible that an intervention on only one or the other, but not both, fails to induce the desired effect. We believe our work sets the stage for follow on studies that seek to identify the circuits which drive planning behaviors, but that due to the high complexity of this endeavor, such work is out of scope for the current manuscript.
>
> **[Logistics complexity]** We can explain why we simplified the Logistics environment. The Logistics environment was simplified  so that fine-tuning the LLMs was easy to do many times over, while keeping sufficient difficulty of the planning task to induce interesting world model recovery behavior. We wanted to use the same environment for both non-State-CoT and State-CoT LLMs, and the extra airplanes and vehicles creates planning problems with excessively large state spaces (# of state predicates) that make training State-CoT LLMs difficult (a key limitation of this approach). More concretely, at the suggestion of Reviewer UzUB, we re-ran our random walk and random walk++ experiments, controlling for dataset size so that these match the optimal plan datasets. In the updated manuscript, we can see in Table 1 that, with equal training data, the Logistics Plansformer LLMs with State-CoT fine-tuned on random walk and random walk++ data attain goal reach rates of only 5% and 2%. Without State-CoT, the goal reach rates are around 100%. We hope this convinces the reviewer that the Logistics domain we use is sufficiently challenging.
>
> We look forward to hearing whether we were able to satisfy the requested changes with our response.

---

### Review · Reviewer_UzUB · 2026-02-11

**Summary Of Contributions:**

The paper studies whether supervised fine-tuning (SFT) on classical planning leads LLMs to “recover a world model,” defined as:
- hidden states that linearly encode action validity and state predicates
- generative behavior that assigns higher probability to valid than invalid actions. Gemma2-9B-Instruct is fine-tuned on Blocksworld and Logistics under three data distributions (optimal plans, random walks, random walk++ for better exploration) plus a State-CoT variant, and evaluated via linear probes and a probability-ranking validity metric, including an OOD scale test (6 blocks).

Strengths:
- Clear framing (representations vs probabilities)
- Thoughtful distribution/State-CoT ablations
- Good evidence for a dissociation where probes can succeed while probability-based validity ranking fails (notably for optimal-plan SFT)
- Appropriately cautious about causal claims.

Weaknesses:
- Comparisons are confounded by large dataset-size/token-budget differences across distributions
- Interpretability analyses are computed only on greedily decoded valid generated plans (invalid plans discarded), narrowing scope
- OOD is limited (scale only, non-State-CoT)
- Probability metric is very strict and needs clearer specification (logits vs log-probs, normalization).

**Audience:**

Yes

**Audience Explanation:**

The paper is relevant to ongoing debates about whether LLMs learn structured internal representations in sequential decision-making settings, and how to interpret the relationship between internal “knowledge” (as probed) and externally observed probabilities/behavior. The work sits at the intersection of LLM planning, interpretability, and dataset design; the empirical dissociation between probeability and probability-based validity discrimination is likely of broad interest to TMLR readers working on mechanistic interpretability, reasoning, and agentic behavior.

**Broader Impact Concerns:**

No direct concerns

**Claims And Evidence:**

Yes

**Claims Explanation:**

Overall, the main empirical claims are supported by the reported experiments under the authors’ definitions of world model recovery. The two definitions are explicitly stated and matched to concrete evaluation procedures.

The evidence for dissociation between internal representation quality and generative validity discrimination is convincing in the reported results: action validity probes reach high F1 in intermediate/upper layers across most settings (on the valid-plan subset), including optimal-plan fine-tuning, whereas probability-based classification can be near-chance/very low for optimal training and markedly higher for random-walk training.

However, two issues temper how strongly the evidence supports the interpretation that “state coverage” (or “exposure to broader valid actions”) is the driver. First, the datasets differ in size across distributions (optimal plans are far fewer than random-walk plans), which introduces a token-budget confound. Second, the analysis pipeline discards invalid generated plans and computes metrics only on valid greedy-decoded plans, meaning the results characterize representations/probabilities on trajectories where the model already behaves correctly. These limitations do not invalidate the core measurements, but they narrow the scope and weaken causal attribution about why different training distributions yield different recovery behaviors.

**Requested Changes:**

- Control token budget / dataset size: match distributions by tokens/updates (or subsample) to reduce confounding.
- Address filtering that only keeps valid plans: discuss selection effects and, if feasible, analyze trajectories around failures (for example, include invalid plans or condition on the first-invalid step).
- Clarify and stress-test the generative metric (Important): specify whether action scores use logits vs log-probs, how they are aggregated across tokens (sum vs mean), and what normalization is applied
- OOD scope (Strengthen): clarify OOD is scale-only (and non-State-CoT); consider at least one additional shift or include State-CoT OOD.

---

> ### Author Response · Authors · 2026-04-12
> **Author response**
>
> Dear Reviewer UzUB,
>
> Thank you for taking the time to review our manuscript and suggest changes that we believe have greatly strengthened the validity of our findings. We have uploaded a revision of the manuscript with changes made in blue. Here we summarize the changes relevant to your review.
>
> **[Control dataset size]** We agree that an imbalance in training dataset size between random walk, random walk++, and optimal datasets potentially confounds the comparisons between LLMs fine-tuned on these datasets. To address this, we downsampled the random walk and random walk++ datasets such that they have the same number of plans as the optimal datasets. We re-trained those LLMs on the new data. Sections 5 and 6 have been updated accordingly with the new downsampling details, performance summary, and world model recovery results, respectively. The world model recovery results were unchanged. The only impacts are summarized here:
>
>   -  The planning performance of some of the State-CoT models dropped. We added the following explanation in Section 5: *"With State-CoT, the LLMs fine-tuned on random walk and random walk++ datasets saw a performance drop compared to optimal plans. State-CoT introduces a non-trivial increase in the length of training sequences, which can increase the amount of training tokens needed to achieve a certain level of performance."* However, this only had a minor impact on the measured State-CoT world model recovery.
>  -  With less data, the OOD generalization performance of the BlocksWorld Plansformer LLM fine-tuned on random walk++ data drops to the level of performance of by the Plansformer fine-tuned on optimal plans. The large jump in OOD performance from random walk to random walk++ is still observed. We added this sentence to Section 6.4: *"Plansformers fine-tuned with random walk++ and optimal data see similar drops in performance when introducing a sixth block, followed by a steady decline as the number of unseen blocks increases, to near zero goal reach rate with ten blocks."*.
>
> **[Filtering for valid plans]** The reviewer correctly observes that we discard invalid plans, e.g., plans where the LLM generates an invalid action at an intermediate plan step. We did this primarily to simplify our analyses, as these invalid plans must be treated with a separate, more complicated analysis procedure than the one we used and described in Section 4.1. We have added the following to our discussion of limitations in Section 8: *"For example, we focused our interpretability analyses on only valid LLM-generated plans, which adds a layer of context to our results that assumes all analyzed plans are ``good'' plans. An immediate follow-up study could try isolating the last valid plan step within an \emph{invalid} plan to analyze world model recovery under a relaxation of this assumption."*
>
> **[Generative metric]** Thanks for pointing out our inconsistent discussion about logits vs. log probs. In our work, we use the average of each action string's token's log probabilities. By taking the mean, our metric becomes insensitive to the number of tokens in the action string, which varies in length. We compute log probabilities by taking a log softmax of the logits. We have clarified references to this throughout the paper, including in Figure 3.
>
> **[OOD Scope]** To strengthen our OOD experiments, and in line with Reviewer Csk6's request, we have added additional OOD plans with 7-10 blocks. Section 6.4 and Table 3 have been updated with the new results. We see no change in world model recovery metrics. The majority of the new text added to Section 6.4 reads: *"We did not evaluate LLMs fine-tuned with State-CoT because their performance on this OOD task is confounded by the challenge of \emph{length generalization}. The number of state tokens that are generated for each plan step inflates the size of the generated plans dramatically; specifically, OOD plans have roughly 2.5$\times$ more tokens than in-distribution plans."* and *"Table 3 shows the planning performance.
> Plansformers fine-tuned with random walk++ and optimal data see similar drops in performance when introducing a sixth block, followed by a steady decline as the number of unseen blocks increases, to near zero goal reach rate with ten blocks."*.
>
> We hope this has adequately responded to your requested changes and look forward to your reply.

---

### Review · Reviewer_GnWA · 2026-03-16

**Summary Of Contributions:**

This article trains LLMs (gemma2-9b) on STRIPS plans, and studies their ability to infer a world model by applying linear probes to representations and scoring generation probabilities. The contribution is an observational study in two domains (Blocksworld, Logistics). It confirms that learned representations capture state predicates and action validity.

While it may not be self-evident that representations align with state predicates, these domains are designed such that predicates are relevant to the action validity, and action validity directly correlates with the probability in the dataset of valid plans, so I find this result unsurprising.

The method measures not if the language model learns **a** world model (which we already know it does by observing it's effective performance), but if it learns **our** world model (the one encoded by state predicates and actions). This nuance is not discussed.
The article also does not convincingly establish why learning our model is crucial. I do like the later argument about (OOD) generalisation, which I agree is the real underlying objective.

**Audience:**

Yes

**Audience Explanation:**

Planning is of great interest to the community, both in itself and due to its model of reasoning in general. Learning world models from observed trajectories is quite a general problem, and identifying if LLMs recover world models in classical planning is a meaningful stepping stone to understanding it.

**Broader Impact Concerns:**

I have no concerns about the impact.

**Claims And Evidence:**

Yes

**Claims Explanation:**

Technically, it appears self-consistent and correct.

Experimental observations are well supported by figures, and figures result from well described experimentation.

For reproducibility, details such as domain depth, epoch numbers and other hyperparameters are given.

**Requested Changes:**

- Section 3.1 §1: The reason for optimal trajectories not being the best source to learn world models from should be reproduced from the cited literature. Limited state space coverage is mentioned in relation to cycles in random walks, but also plays a large role in optimal trajectories. However, a tradeoff may need to be made, and I would have expected the authors to align the world model coverage of interest with the domain, and making it robust (eg random deviations from the optimal path may be needed for deep sequential domains).

- The authors do very well frame OOD generalisation as an objective, and I agree with that. However, there is some conceptual dissonance whether recovering a good world model is an objective or not. Overall, it seems not to be an objective but rather an observational metric. But the last paragraph on page 12 uses 'best results' and 'edging out...' which suggests it is then seen as one. It would be good to keep a consistent message throughout.

Minor comments
- Section 3, §2: The authors' concerns of STRIPS being unnatural is maybe unwarranted. LLMs are trained on vast data, including code, and some directly support structured generation of constrained outputs (like masking valid tokens for JSON, markdown etc). The authors could strengthen their argument in this light.
- Figure 4: Layer axis is labelled {0, 1}, so the axis name should be changed to 'Layer depth (fraction #/N)'.

- The LLM does not need to represent all facts; just the necessary ones. It is unclear from the current exposition in how far the tested state facts were required for the task at hand. If they are not, there is no reason to expect them to be represented. This should be explicitly addressed or at least discussed.

---

> ### Author Response · Authors · 2026-03-24
> **Question for Reviewer GnWA**
>
> Dear GnWA,
>
> Thank you for your time spent reading our manuscript and providing thoughtful comments. As we work to make requested changes, we have one clarification question for you.
>
> > The method measures not if the language model learns a world model (which we already know it does by observing it's effective performance), but if it learns our world model (the one encoded by state predicates and actions).
>
> The reviewer makes an interesting point that our use of the term “world model” makes an implicit assumption that a “world model” is a *human* world model, i.e., one that makes sense to a human, e.g., one that makes use of concepts, objects, properties, and relations. *We ask, in reply, what would be meant by a different, non-human “world model”?*
>
> If it is just “whatever is going on inside that makes the network (usually) arrive at the right answer”, we argue that this is too general a notion to be actionable.  Moreover, explaining the actions of neural networks at a mechanistic level somewhere between human interpretable concepts and the matrices and vectors that constitute the core network (e.g., circuit identification) is an interesting endeavor, but we believe this is outside the scope of the current work.

---

> ### Author Response · Authors · 2026-04-12
> **Author response**
>
> Dear Reviewer GnWA,
>
> Thank you for your review of our manuscript. We have uploaded a new revision with changes highlighted in blue text. Here, we summarize the changes in response to your comments:
>
> **[State space coverage]** In Section 3.1 we added an explanation on why optimal trajectories may be sub-optimal for world model recovery *"Evidence suggests this is because optimal action sequences encourage learning representations that only help predict \emph{strategically good actions}, rather than \emph{all valid actions} (e.g., all next legal moves from the current board game state)~\citep{li2022emergent}"*. We also investigated a naive strategy to enhance the state space coverage of the optimal dataset, where we mixed a small fraction (10%) of the enhanced random walk data with 90% of the optimal data for fine-tuning. This had no noticeable change on the plan performance (Table 1) or action validity linear probe results (Figure 4); however, we observed a *drop* in valid action classification performance (Table 2, Figure 6). Consequentially, we did not add these new results to the manuscript. As the reviewer points out, our paper frames world model recovery as an observational metric, not an objective of the paper. Therefore, we are inclined to leave deeper exploration of fine-tuning data mix (e.g., crafting the right amount of random perturbations to optimal plans) to future work.
>
> **[Framing of world model recovery as observational metric]** Thank you for highlighting this inconsistency. We have adjusted the language we use to refer to world model recovery through Section 6 with a more neutral tone. We maintain the language used to compare OOD generalization performance across fine-tuned LLMs, emphasizing that we see it as an objective to improve.
>
> **[STRIPS]** Thanks for pointing this out. We agree. We removed the sentence about the unnatural nature of STRIPS.
>
> **[Clarification about probed state predicates]** To clarify that we use plan-specific predicate labels for training and testing state predicate probes, we have added the following sentence to Section 6.2 *"However, we do not train or evaluate any state probes that, for a given plan, correspond to an irrelevant predicate. For example, for a BlocksWorld problem with only 4 blocks, we do not probe for grounded state predicates that have \texttt{block5}"*
>
> Figure 4, 5, and 8's layer axis has been fixed.
>
> We hope we have adequately addressed your requested changes and comments, and look forward to your reply.

---

### Author Response · Authors · 2026-04-12
**Updated manuscript summary**

We thank the reviewers and action editor for helping us improve our manuscript. We have uploaded a new version of the manuscript with requested changes made in blue text. Here, we summarize the main changes:

**Re-trained and evaluated random walk and random walk++ models**:  Random walk and random walk++ datasets were downsampled to match the number of optimal plan examples, improving fairness of comparisons and controlling for a potential confounder. This required us to recompute all of the corresponding world model recovery metrics in Section 6. However, these results were unchanged. The new findings are:

   * The planning performance of some of the State-CoT models dropped. We added the following explanation in Section 5: *"With State-CoT, the LLMs fine-tuned on random walk and random walk++ datasets saw a performance drop compared to optimal plans. State-CoT introduces a non-trivial increase in the length of training sequences, which can increase the amount of training tokens needed to achieve a certain level of performance."* However, this only had a minor impact on the measured State-CoT world model recovery.
  * With less data, the OOD generalization performance of the BlocksWorld Plansformer LLM fine-tuned on random walk++ data drops to the level of performance of by the Plansformer fine-tuned on optimal plans. The large jump in OOD performance from random walk to random walk++ is still observed. We added this sentence to Section 6.4: *"Plansformers fine-tuned with random walk++ and optimal data see similar drops in performance when introducing a sixth block, followed by a steady decline as the number of unseen blocks increases, to near zero goal reach rate with ten blocks.".*

**Added motivation for our training data comparison**, explaining that optimal plans may teach the model to favor strategically good actions rather than representing all valid actions.

**Made metric definitions more precise by:**

  * Renaming the extraction subsection to include token log probabilities, specifying that these are computed as the log-softmax of logits, and clarifying that action scores use average token log probabilities. We also updated Figure 3 and its caption accordingly.
  * Justified the probe evaluation metric by adding a detailed explanation for using F1 under heavy class imbalance, where invalid actions (negative class) are the large majority.

**Clarified state-probe methodology** by noting that we do not probe irrelevant predicates that involve objects absent from a given instance (e.g., we don't probe for `block5` in a 4-block problem)

**Expanded the OOD setup and caveats by:**

  * Including instances with 7-10 blocks from PlanBench for a total of 261 instances. No change in world model recovery metrics observed.
  * Explaining why State-CoT OOD was not evaluated: it is confounded by length generalization, with OOD plans having about 2.5× more tokens
  * Clarifying the different OOD degradation patterns for random walk++, optimal, and random walk models as the number of held out blocks increase from 6 to 10.

**Added a limitation/future-work point:** Our interpretability analysis only considers valid generated plans, and a follow-up could study the last valid step inside invalid plans.